# Resistance Exercise Training Attenuates the Loss of Endogenous GLP-1 Receptor in the Hypothalamus of Type 2 Diabetic Rats

**DOI:** 10.3390/ijerph16050830

**Published:** 2019-03-07

**Authors:** Se Hwan Park, Jin Hwan Yoon, Dae Yun Seo, Tae Nyun Kim, Jeong Rim Ko, Jin Han

**Affiliations:** 1Institute of Sports Medicine, Hannam University, Daejeon 34430, Korea; psh8179@gmail.com (S.H.P.); yoonjh@hannam.ac.kr (J.H.Y.); 2National Research Laboratory for Mitochondrial Signaling, Department of Physiology, BK21 Plus Project Team, College of Medicine, Cardiovascular and Metabolic Disease Center, Inje University, Busan 47392, Korea; sdy925@gmail.com (D.Y.S.); kimtn031@gmail.com (T.N.K.); kjrsos0217@gmail.com (J.R.K.); 3Department of Health Science and Technology, Graduate School, Inje University, Busan 47392, Korea

**Keywords:** type 2 diabetes, resistance exercise, hypothalamus, glucagon-like pepetide-1 receptor, glucose transporter 2

## Abstract

The aim of this study was to investigate the effects of resistance exercise training on hypothalamic GLP-1R levels and its related signaling mechanisms in T2DM. The animals were separated into three groups: a non-diabetic control (CON), diabetic control (DM), and diabetic with resistance exercise (DM + EXE) group. The resistance exercise training group performed ladder climbing (eight repetitions, three days per week for 12 weeks). Body weight was slightly lower in the DM + EXE group than the DM group, but difference between the groups was not significant. Food intake and glucose were significantly lower in the DM + EXE group than in the DM group. The blood insulin concentration was significantly higher and glucagon was significantly lower in the DM + EXE group. The DM + EXE group in the hypothalamus showed significant increases in GLP-1R mRNA, protein kinase A (PKA), glucose transporter 2 (GLUT2), and protein kinase B (AKT) and significant decrease in protein kinase C-iota (PKC-iota). Antioxidant enzymes and apoptosis factors were significantly improved in the DM + EXE group compared with the DM group in the hypothalamus. The results suggest that resistance exercise contributes to improvements the overall health of the brain in diabetic conditions.

## 1. Introduction

Type 2 diabetes mellitus (T2DM), the most common type of diabetes, accounts for more than 90% of patients with diabetes and is rapidly emerging as one of the most serious health problems worldwide [1]. Progressive hyperglycemia, a major feature of T2DM, contributes to the dysfunction of several organs, including the central and peripheral systems, eventually leading to diabetic complications [2,3]. Many patients with T2DM are at an elevated risk of various complications due to a lack of appropriate glycemic control. For these reasons, the importance of glycemic control via diet, medication, and exercise has been emphasized [4,5,6,7]. 

Normal glycemic control is mediated by multiple mechanisms involving the central and peripheral systems. Recently, the role of the brain in peripheral energy and glucose homeostasis has been proposed [8]. The hypothalamus, in particular, is the primary regulatory region for food intake and is directly related to blood glucose levels, which are also involved in hormone signaling, such as insulin and leptin [9]. However, the brain is a highly sensitive and vulnerable tissue influenced by chronic hyperglycemia, which can cause the gradual impairment of normal brain functions.

Pharmacologically, incretin hormone-based therapies are widely used to improve hyperglycemia in patients with T2DM [10]. Glucagon-like peptide-1 (GLP-1) is major incretin hormone and is secreted from L-cells in response to nutrient ingestion. The physiological activity of GLP-1 is activated by binding to its receptor (GLP-1R), which increases glucose-dependent insulin secretion and has many extra-pancreatic effects, such as glucose lowering, anti-inflammatory and gastric emptying reduction effects as well as the inhibition of food intake [11,12]. GLP-1R is expressed in various regions of the central nervous system (CNS) [13]. GLP-1R activation within the hypothalamus increases insulin secretion in ß-cells and decreases hepatic glucose production. GLP-1R-mediated intracellular signaling augments cyclic adenosine monophosphate (cAMP) levels and promotes the protein kinase A (PKA)/exchange protein directly activated by cAMP 2 (Epac2) pathway [14,15]. Additionally, the GLP-1R-mediated regulation of adenosine monophosphate-activated protein kinase (AMPK) is associated with feeding behavior and body weight gain. The loss of glucose sensing by decreased glucose transporter-2 (GLUT-2) leads to the improper activity of AMPK in the hypothalamus [16,17].

Recent research has focused on the development of pharmacological agents, such as GLP-1R agonists (GLP-1RAs) and dipeptidyl-peptidase IV (DPP-IV) inhibitors; these drugs have well-documented effects based on studies of humans and rodents. For example, the once-daily administration of liraglutide is an effective strategy for altering HbA1c levels and body weight loss in obese patients with T2DM [18]. The administration of GLP-1RA to the hypothalamus and hindbrain reduces food intake and improves glucose tolerance [19,20]. Moreover, hypothalamic neuronal activity is increased by GLP-1RA [21,22]. However, despite the effectiveness of incretin-based drugs, most patients with T2DM do not reach normal blood glucose levels and side effects need to be considered. Growing evidence suggests that hyperglycemia consistently decreases endogenous GLP-1R levels, which may reduce the efficacy of GLP-1RA therapies [23,24]. Based on these reports, it is necessary to develop new approaches to safely treat hyperglycemia as well as to preserve endogenous GLP-1R.

Exercise, a lifestyle factor, is central to the management of T2DM because it helps to treat the related body weight and blood glucose control abnormalities. Likewise, exercise has many health benefits as a non-pharmacological method in T2DM and the importance of exercise initiation at the early stage is well established. It is generally accepted that aerobic exercise training restores T2DM-induced dysfunctions in several organs and tissues. Many recent studies have proposed that resistance exercise training has similar effects to those of aerobic exercise training, including positive effects on obesity, hypertension, and T2DM [25,26]. If exercise in the early stage of the T2DM can delay the progressive loss of endogenous GLP-1R levels, it may be a novel strategy for glycemic lowering prior to pharmacological treatments and improving the effects of incretin treatment.

The aim of this study was to investigate that effects of resistance exercise training on hypothalamic GLP-1R levels and its related signaling mechanisms in T2DM. We hypothesize that resistance exercise training in the early stage of T2DM may contribute to glycemic management by attenuating endogenous GLP-1R losses and increasing glucose sensing in the hypothalamus.

## 2. Materials and Methods 

### 2.1. Experimental Animals

Sixteen male Otsuka Long-Evans Tokushima Fatty (OLETF) rats were randomly assigned to diabetic control (DM, *n* = 8) and diabetic exercise-trained groups (DM + EXE, *n* = 8) at 22 weeks of age. Long-Evans Tokushima Otsuka (LETO) rats were used as the non-diabetic control (CON, *n* = 8). OLETF rats are characterized by a mutated cholestykinin-1 receptor, resulting in a hyperphagic phenotype; they are an established model of obesity, insulin resistance, and T2DM [27]. Animals were housed at two rats per cage in a temperature-controlled (20° ± 2.5 °C) and light-controlled (12:12 h light-dark diurnal cycle) room. All animal experiment procedures were approved by the Institutional Animal Use and Care Committee of Hannam University (HNU2016-16). 

### 2.2. Exercise Protocol

The resistance training protocol was adapted from a previous study [28]. Rats in the exercise-trained group (DM + EXE) were trained to climb a 135-cm ladder (85 °C incline) with weight secured to their tails. For 12 weeks, the rats were subjected to one training session per day for 3 days/week. In the first week, the rats were familiarized with climbing to the top of the cage without a weight on their tails. After 1 week of adaptation, training sessions were commenced with an intensity of 30% of each rat’s body weight; an angling weight was attached to the tail with a plastic hairclip and string. Rats began climbing from the bottom of the ladder and were forced to climb to the top. When they reached the top, 1 min of rest was provided and the next trial was initiated. Subsequent trials were started from the bottom, and 15 g was added to the prior weight at every trial. If a rat was able to climb 8 times with increasing weights, the training session was complete. Schematic of experiment design is shown Figure 1.

### 2.3. Tissue Preparation and Blood Parameters

Rats were anesthetized by the Intraperioneal (IP) injection of Zoletil 50 (10 mg/kg i.p.; Vibac Laboratories, Carros, France). Tissue samples were collected from the hypothalamus, frozen on ice, and stored at −80 °C until use. Samples were homogenized using RIPA buffer. Samples were spun at 14,000 rpm for 15 min at 4 °C, and the total protein concentration of the supernatant was determined by a Bradford assay. The concentrations of blood glucose, insulin, and glucagon were measured using enzyme-linked immunosorbent assay kits. Blood samples were taken from the heart and were drawn into heparinized tubes. The heparinized tubes were centrifuged for 10 min at 10,000× g to obtain plasma samples. 

### 2.4. Western Blotting

The hypothalamus was homogenized in lysis buffer (50 mM HEPES, 10 mM EDTA, 100 mM NaF, 50 mM sodium pyrophosphate, 10 mM sodium orthovanadate, and 1% Triton at pH 7.4) supplemented with protease/phosphatase inhibitor cocktails (Thermo Fisher Scientific, Waltham, MA, USA). The protein concentration was determined by the BCA method (BCA Protein Assay Kit, Thermo Fisher Scientific, Waltham, MA, USA). For the protein assay, equal amounts of protein (20 µg) were electrophoresed on 8–10% SDS-PAGE gels and transferred to a nitrocellulose membrane. The membranes were blocked with 5% non-fat milk powder in TBST buffer and incubated overnight at 4 °C with the primary antibody. GLUT2 (Abcam, Cambridge, UK), AMPK, PKC-iota, AKT, PKA, Epac2 (Cell Signaling Technology, Danvers, MA, USA), superoxide dismutase 1 (SOD1), SOD2, Bax, Bcl-2, Caspase-3 (Santa Cruz Biotechnology, Santa Cruz, CA, USA), and GAPDH (Abcam, Cambridge, UK) were used at a dilution of 1:1000. The blots were visualized by Super Signal West Pico Chemiluminescent Substrate (Thermo Fisher Scientific, Waltham, MA, USA) and quantified by densitometry using Image J (NIH, Bethesda, MD, USA).

### 2.5. Real-Time PCR

Total RNA from the hypothalamus was extracted with TRIzol (Invitrogen, Cergy Pontoise, France), and single-stranded cDNA was synthesized from 10 µg of total RNA using random hexamer primers (Applied Biosystems, Courtaboeuf, France). Real-time RT-PCR was performed to measure GLP-1 receptor levels, as previously described [29]. The following primers were used: GLP-1R forward primer 5’-GAA GAG TCC AAG CAA GGA GA-3’, reverse primer 5’-GAC CAA GGC AGA GAA AGA AA-3’, GAPDH forward primer 5’-CAG GAG CGA GAT CCC GC-3’, reverse primer 5’-CCT TTT GGC CCC ACC CT-3’.

### 2.6. Statistics

Statistical analyses were performed using SPSS 23.0 (SPSS, Inc., Chicago, IL, USA). All data are presented as means ± standard deviation. Statistical differences were determined by one-way ANOVA with Duncan post hoc tests. The level of significance was set to *p* < 0.05.

## 3. Results

### 3.1. Resistance Exercise Improved Food Intake, Gluocse, Insulin, and Glucagon Levels in T2DM Rats Regardless of Body Weight

Body weight was slightly decreased in the DM + EXE group compared to the DM group, but the difference between the two groups was not significant (Figure 2A). Average food intake (g/week) significantly decreased by in the DM + EXE group compared with in the DM group (*p* < 0.05, Figure 2B). 

Blood glucose was significantly lower (21%) in the DM + EXE group than in the DM group (*p* < 0.05, Figure 2C). The blood insulin concentration was significantly higher (60%) in the DM + EXE group than in the DM group (*p* < 0.05, Figure 2D) whereas the blood glucagon concentration was significantly lower (29%) in the DM + EXE group than in the DM group (*p* < 0.05, Figure 2E). 

### 3.2. Resistance Exercise Increased GLP-1R Levels by Decreasing PKC-iota and Enhancing the Related Signaling Pathway in the Hypothalamus

The effects of resistance exercise training on GLP-1R were measured by real-time PCR. GLP-1R mRNA levels in the hypothalamus were significantly 85% higher in the DM + EXE group than in the DM group (*p* < 0.05, Figure 3A). PKC-iota protein expression was measured by western blotting. PKC-iota levels were significantly decreased by 48% in the DM + EXE group compared with in the DM group (*p* < 0.05, Figure 3B). PKA levels were significantly higher (by 38%) in the DM + EXE group than in the DM group (*p* < 0.05, Figure 3C) and AKT phosphorylation was significantly increased (by 16%) in the DM + EXE group compared with that in the DM group (*p* < 0.05, Figure 3F). Epac2 levels were slightly increased in the DM + EXE group compared to the DM group, but no significant (Figure 3E).

### 3.3. Resistance Exercise Increased GLUT-2 and Attenuated Increases in AMPK in the Hypothalamus

GLUT-2 protein levels were 25% higher in the DM + EXE group than in the DM group (*p* < 0.05, Figure 4B). We also measured AMPK protein levels by GLUT-2 activation. AMPK phosphorylation was significantly decreased by 44% in the DM + EXE group compared with in the DM group (*p* < 0.05, Figure 4C).

### 3.4. Resistance Exercise Improved Antioxidant Enzyme and Apoptotic Protein Levels

SOD1 protein levels were 57% higher in the DM + EXE group than in the DM group (*p* < 0.05, Figure 5B). SOD2 levels were slightly higher in the DM + EXE group than in the DM group, but the difference was not significant (Figure 5C). Caspase-3 and Bax protein levels were significantly decreased (by 35%, 18%) in the DM + EXE group compared with in the DM group (*p* < 0.05, Figure 5D, Figure 5E), and Bcl-2 protein levels were 40% higher in the DM + EXE group than in the DM group (*p* < 0.05, Figure 5F).

## 4. Discussion

Incretin hormone-based therapies are commonly used on patients with T2DM to reduce blood glucose levels. However, it may require a more efficient and safer strategies for hyperglycemia without side effects. In this study, we demonstrated that food intake, glucose, and glucagon levels were higher and insulin secretion was lower in the DM group than in the CON and DM + EXE groups. Furthermore, decreased GLP-1R mRNA levels were related to PKC-iota activation, and antioxidant enzyme and anti-apoptosis protein levels decreased. Additionally, GLUT-2 and AKT protein levels decreased and AMPK protein activity increased in the hypothalamus of T2DM rats. These factors were significantly improved by resistance exercise intervention. Taken together, our findings suggest that resistance exercise training in the early stage of T2DM has a positive effect on glycemic-related factors in the hypothalamus. 

Poor glycemic control is a major barrier to the effective management of T2DM. Reductions in body weight gain and in food intake have been emphasized as strategies to control blood glucose in the T2DM state. According to previous studies, weight loss-induced glucose lowering can delay progressive metabolic degeneration in the early diabetic state [30], and Kim et al. [31] observed that body weight is reduced more substantially in zucker diabetic with exercise (ZDF-EXE) rats than in zucker diabetic control rats after eight weeks of resistance exercise training. Our results are in close agreement with theses previous findings. In the present study, body weight was slightly lower in the DM + EXE group than in the DM group, whereas food intake and blood glucose were significantly different. Moreover, resistance exercise training effectively increased insulin secretion and decreased glucagon production (Figure 2). 

The hypothalamus plays a pivotal role in controlling energy homeostasis as well as glucose metabolism, affecting insulin and glucagon secretion via direct and indirect pathways [32]. GLP-1R is expressed in several cells and organs, including the hypothalamus, kidney, lung, heart, endothelial cells, neurons, and pancreas [33]. In particular, the hypothalamus integrates and conveys a variety of signals to regulate glucose homeostasis. In T2DM conditions, hypothalamus dysfunction is caused by hyperglycemia, which can impair signaling pathways involved in overall metabolic processes. Hyperglycemia increases oxidative stress, cell damage, and apoptosis and inhibits various cell surface receptors. Kim et al. [34] reported that GLP-1R levels are decreased in high glucose-treated retinal pigment epithelium. Xu et al. [35] reported that high glucose-activated PKC reduces GLP-1R expression in the islets of hyperglycemic rats. Consistent with the findings of previous studies, we observed that GLP-1R mRNA levels were reduced by increased PKC-iota and PKA and AKT levels were significantly decreased in the DM group. However, resistance exercise training for 12 weeks significantly increased GLP-1R mRNA (Figure 3A) as well as PKA (Figure 3C) and AKT (Figure 3F) protein levels by altering PKC-iota expression (Figure 3B). Exercise is essential for the management of T2DM and it may have a positive effect on GLP-1R signaling in the CNS. However, the effects of exercise itself on hyperglycemia-induced endogenous GLP-1R levels in the hypothalamus are unknown.

To the best of our knowledge, this is the first evaluation of endogenous GLP-1R mRNA levels and the related signaling pathway in the hypothalamus in response to regular exercise. Similar to our study, Mensberg et al. [36] reported that the combination of regular exercise and GLP-1RA (liraglutide) for hyperglycemia management leads to normal HbA1C levels and decreases body weight gain. Exercise improves the effectiveness of GLP-1RA therapy by enhancing β-cell function, which is directly linked to the hypothalamus [37]. Furthermore, various exercise types can increase the blood GLP-1 concentration [38,39]. It is generally accepted that the enhancement of the activity of the GLP-1/GLP-1R system in the hypothalamus is useful for the treatment of T2DM [40]. As a treatment strategy for endogenous receptor loss, GLP-1RAs improve insulin secretion, glucose reduction, and appetite control and have anti-inflammatory effects [41,42]. In view of this point, the inhibition of endogenous GLP-1R loss may be important. We suggest that the inhibition GLP-1R loss by regular exercise from the initial diagnosis of T2DM is important, before pharmacological treatment. 

In addition, similar to in pancreatic β-cells, GLUT-2 expression in the brain is involved in sensing changes in the blood glucose concentration [43,44]. Glucose sensing in appetite centers is involved in feeding behavior. Hypothalamic glucose sensors via glucose transporters play an important role in controlling insulin secretion from β-cells, and neuronal activation within the hypothalamus leads to appropriate food intake. The loss of glucose sensing by decreased GLUT-2 leads to the improper activity of AMPK in the hypothalamus [45]. The activation of hypothalamic AMPK increases food intake, whereas its inhibition decreases food intake. Additionally, decreased GLUT-2 expression is associated with a disruption in the secretion of appetite hormones, such as GLP-1, glucose-dependent insulinotropic polypeptide (GIP), and peptide YY (PYY) [46]. GLUT-2-mediated AMPK signaling in the hypothalamus is involved in energy homeostasis, resulting in improved glycemic control [47]. In the present study, we observed that resistance exercise training significantly increases GLUT-2 and decreased AMPK (Figure 4). Other studies have reported that exercise training improves GLUT-2 and AMPK expression in several peripheral tissues [48,49,50].

Moreover, T2DM-mediated brain dysfunction is associated with inflammation, antioxidation, and apoptosis. We found that resistance exercise training increased SOD1 and Bcl-2 protein levels, and decreased Bax and caspase-3 protein levels (Figure 5), which may contribute in part to GLP-1R activation. Exercise-induced antioxidant activity offsets apoptosis and synapse damage in the CNS caused by oxidative stress [51]. This current study has a limitation as follows; we did not measure body composition, adipose tissue, lipid profile, and muscle mass.

## 5. Conclusions

Regular exercise prevents the development of diabetes-related complications by decreasing chronic hyperglycemia. The results of this study suggest that regular resistance exercise training may contribute to improvements in insulin secretion, glycemic lowering, and glucagon inhibition by the delay of endogenous GLP-1R loss in the CNS in initial T2DM. Schematic diagram of the mechanisms for the resistance exercise training in hypothalamus of T2DM rats is shown Figure 6. However, further studies are needed to understand the underlying mechanism.

## Figures and Tables

**Figure 1 ijerph-16-00830-f001:**
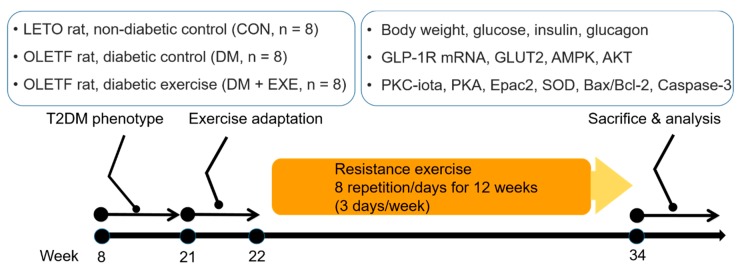
Schematic of experimental design. T2DM: type 2 diabetes mellitus; LETO: Long-Evans Tokushima Otsuka; OLETF: Otsuka Long-Evans Tokushima Fatty; GLP-1R: Glucagon-like peptide-1 receptor; GLUT2: glucose transporter 2; AMPK: adenosine monophosphate-activated protein kinase; AKT: protein kinase B; PKC-iota: protein kinase C-iota; PKA: protein kinase A; Epac2: exchange protein directly activated by cAMP 2; SOD: superoxide dismutase; Bax: BCL2-associated X protein; Bcl-2:. B-cell lymphoma 2.

**Figure 2 ijerph-16-00830-f002:**
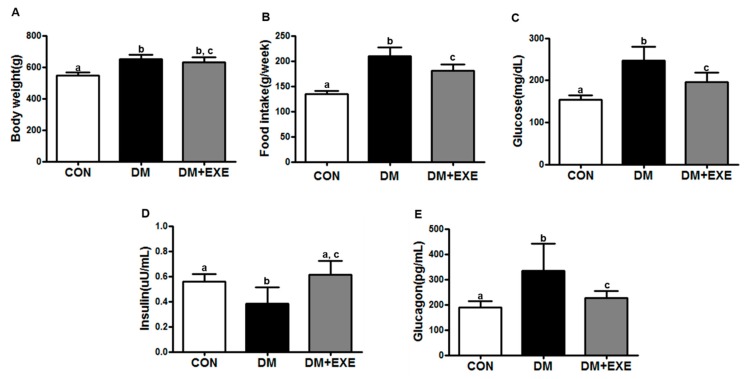
Resistance exercise training decreased food intake, glucose, and glucagon levels and increased insulin levels in T2DM rats. All data are presented as the Mean ± SD. (**A**) body weight, (**B**) food intake, (**C**) glucose, (**D**) insulin, (**E**) glucagon. CON: non-diabetic control group; DM: diabetic control group; DM + EXE: diabetic with resistance exercise training group. ^a, b, c^ Different letters indicate a significant difference between groups (*p* < 0.05).

**Figure 3 ijerph-16-00830-f003:**
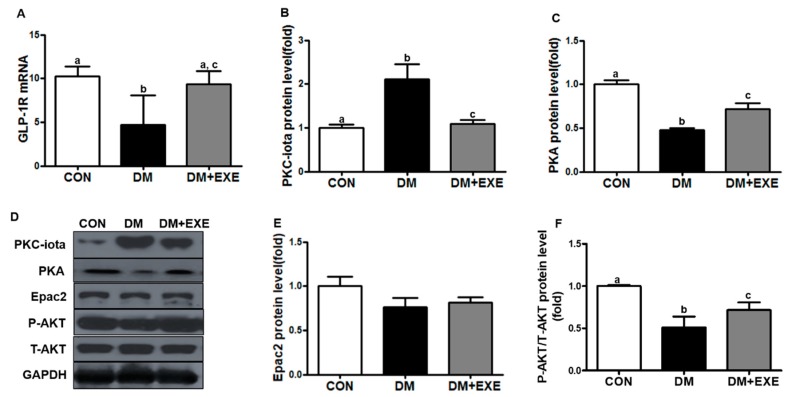
Resistance exercise training inhibited loss of GLP-1R and enhanced signaling pathway in the hypothalamus of T2DM rats by decreasing PKC-iota level. All data are presented as the Mean ± SD. (**A**) glucagon-like peptide-1 receptor mRNA (GLP-1R mRNA), (**B**) protein kinase C-iota (PKC-iota), (**C**) protein kinase A (PKA), (**D**) representative western blotting band, (**E**) exchange protein directly activated by cAMP 2 (Epac2), (**F**) phosphorylation protein kinase B/total protein kinase B (P-AKT/T-AKT). CON: non-diabetic control group; DM: diabetic control group; DM + EXE: diabetic with resistance exercise training group. ^a, b, c^ Different letters indicate a significant difference between groups (*p* < 0.05).

**Figure 4 ijerph-16-00830-f004:**
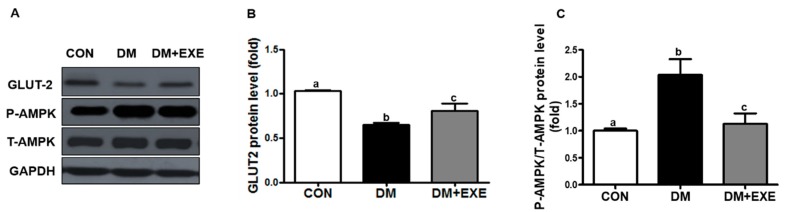
Resistance exercise training increased GLUT2 and attenuated AMPK protein levels in the hypothalamus of T2DM rats. All data are presented as the Mean ± SD. (**A**) representative western blotting band, (**B**) glucose transporter 2 (GLUT2), (**C**) phosphorylation-adenosine monophosphate-activated protein kinase (P-AMPK)/total-adenosine monophosphate-activated protein kinase (T-AMPK). CON: non-diabetic control group; DM: diabetic control group; DM + EXE: diabetic with resistance exercise training group. ^a, b, c^ Different letters indicate a significant difference between groups (*p* < 0.05).

**Figure 5 ijerph-16-00830-f005:**
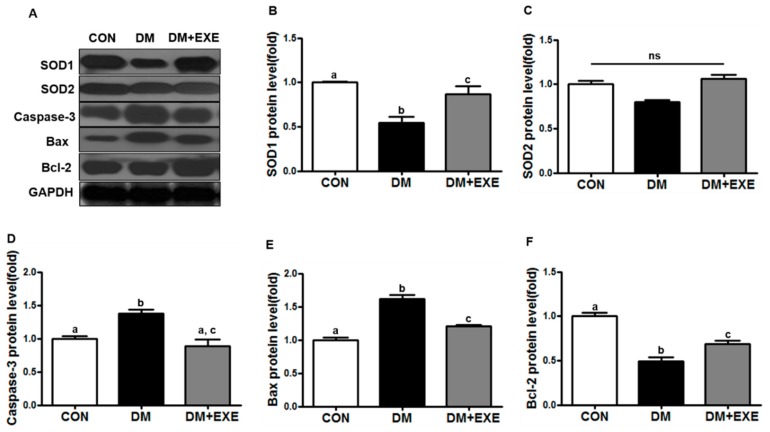
Resistance exercise training improved antioxidant enzymes and apoptosis protein levels such as, caspase-3 and Bax/Bcl-2 in the hypothalamus of T2DM rats. All data are presented as the Mean±SD. (**A**) representative western blotting band, (**B**) superoxide dismutase 1 (SOD1), (**C**) superoxide dismutase 2 (SOD2), (**D**) Caspase 3, (**E**) BCL2-associated X protein (Bax), (**F**) B-cell lymphoma 2 (Bcl-2). CON: non-diabetic control group; DM: diabetic control group; DM + EXE: diabetic with resistance exercise training group. ^a, b, c^ Different letters indicate a significant difference between groups (*p* < 0.05). ns: not significant.

**Figure 6 ijerph-16-00830-f006:**
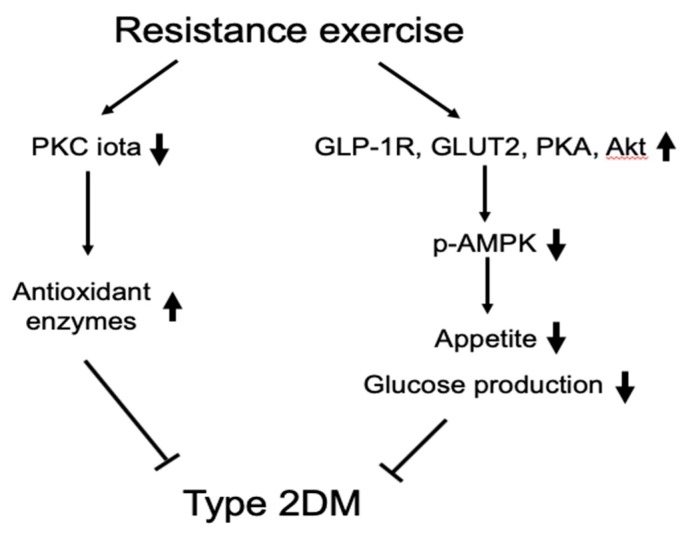
Schematic diagram of the mechanisms for the resistance exercise in hypothalamus of type 2 diabetic rats.

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
