# Peer review of "Resistance Exercise Training Attenuates the Loss of Endogenous GLP-1 Receptor in the Hypothalamus of Type 2 Diabetic Rats"

_ijerph, 2019, doi:10.3390/ijerph16050830_

Round 1

Reviewer 1 Report

In this manuscript, the authors described that resistance exercise training improves the hyperglycemia, increases the insulin secretion and decreases the glucagon by attenuating the GLP-1R loss in the hypothalamus of T2DM rats. Proteins related to glucose metabolism, antioxidant and anti-apoptosis are also activated by resistance exercise training. The study is scientifically sounds, well designed and well written. I have only few comments aimed at improving the quality of the manuscript:

1. The authors used “fasting” to indicate the fasting blood glucose level, however, it will be confused with the fasting insulin level. Please modify it clearly.

2. Figure 1: How about the adipose tissue distribution and the plasma lipid profile?

3. It is well known that exercise/physical activity is the basic management care for T2DM patients. Skeletal muscle is one of the major target organs of insulin actions and plays an essential role in insulin-induced glucose uptake. In this manuscript, the observed effect of resistance exercise training are probably also related to an increased skeletal muscle insulin action. The authors should discuss this point in the discussion section.

4. I would recommend to include a figure/diagram to recapitulate the mechanism by which resistance exercise training improves diabetes. This will be useful for understanding.

Author Response

Thank you for your comprehensive review of our paper. We provided a point-by-point response to your comments below.

Response to Reviewer 1 Comments

Thank you for your comprehensive review of our paper. In line with your comments, we have made the following modifications in the revised manuscript.

Point 1: The authors used “fasting” to indicate the fasting blood glucose level. However, it will be confused with the fasting insulin level. Please modify it clearly.

Response 1: We apologize for the error and thank you for this suggestion. In the revision, we have deleted the word “fasting” and corrected it with blood glucose (Page 4, line 144, 149, 154, Page 4, figure legend).

Point 2: Figure 1: How about the adipose tissue distribution and the plasma lipid profile?

Response 2: Thank you for your helpful comments. Unfortunately, we didn’t measure the adipose tissue and lipid profile in this study. These factors are a limitation in our study (Page 7, lines 256-257). Therefore, if possible, we will reflect these recommended factors in further study.

Point 3: It is well known that exercise/physical activity is the basic management care for T2DM patients. Skeletal muscle is one of the major target organs of insulin actions and plays an essential role in insulin-induced glucose uptake. In this manuscript, the observed effect of resistance exercise training are probably also related to an increased skeletal muscle insulin action. The authors should discuss this point in the discussion section.

Response 3: Thank you for your contractive and comprehensive review of paper. In many studies, it has been demonstrated that resistance exercise decreases adipose tissue and increases muscle mass, which improves insulin-induced glucose uptake in skeletal muscle. We also agree with your comments. However, in recent studies, the role of the brain on peripheral glucose homeostasis has been even more proposed (Schwartz et al. 2013; Jordan et al. 2010). Accordingly, we focused on a direct effect of resistance exercise on GLP-1R and its-related signalling in brain. Although GLP-1R activation increases glucose-dependent insulin secretion, which may contribute to improvements of glucose and energy homeostasis in peripheral including skeletal muscle, we didn’t investigate the effect of resistance exercise training on skeletal muscle insulin action. Therefore, we did not your recommendation in the discussion section. We ask for your understanding regarding this matter. If possible, we will reflect these recommended factors in further study.

Point 4: I would recommend to include a figure/diagram to recapitulate the mechanism by which resistance exercise training improves diabetes. This will be useful for understanding.

Response 4: Thank you for helpful comment. As for the above suggestion, the revised manuscript has provided summarized figures on mechanism induced by resistance exercise training in diabetes (Page 3, lines 106-108; Page 8, lines 264-267).

Figure 1.  Schematic of experimental design 

Figure 6. Schematic diagram of the mechanisms for the resistance exercise in hypothalamus of type 2 diabetic rats.

Reference

1. Schwartz MW1, Seeley RJ, Tschöp MH, Woods SC, Morton GJ, Myers MG, D'Alessio D. Cooperation between brain and islet in glucose homeostasis and diabetes. Nature. 2013 503(7474):59-66. doi: 10.1038/nature12709.

2. Jordan SD, Könner AC, Brüning JC. Sensing the fuels: glucose and lipid signaling in the CNS controlling energy homeostasis. Cell Mol Life Sci. 2010, 67(19):3255-73.

We have shown our responses to your comments in red in the revised manuscript.

Reviewer 2 Report

Park and colleagues have completed experiments examining the relationship between resistance exercise training, GLP1 receptor abundance in the hypothalamus, and several parameters related to diabetes/metabolic syndrome. They have utilized OLETF rats as treatment cohorts with the LETO rat as a non-diabetic control. 

The experiments appear to have been carefully conducted and the experimental design is well suited to evaluate the research questions proposed. The data reported will be of interest to the readership of the journal and offers new insights into an interesting mechanism underlying the benefits of resistance exercise in ameliorating T2DM. The authors are to be commended for examining the effect of resistance training in a rodent model, which is an exercise paradigm that is understudied in the context of metabolic disease in animal models in general.

The manuscript could be strengthened by addressing the points raised below:

For the exercise protocol, is there a way to express the load as a percentage of the animals' maximal capacity? In humans, exercise intensity is usually prescribed on the basis of a percentage of 1 repetition maximum. It would be of interest to know the intensity at which the animals were training.

The authors should change "improved" in reference to food intake in Figure 1B to "decreased." This would make for a more straightforward interpretation of the data.

The authors show in figure 2 that fasting glucose in the trained animals was partially restored to a level close to the controls. However, insulin also was altered by the treatment. Given that both fasting glucose and insulin are important to understanding the whole-body glucose homeostasis of the animals, the authors should include a panel that incorporates both measures in an index such as HOMA-IR or QUICKI, which has been shown to correlate well with insulin clamp studies.

For the WB data on AKT and AMPK, the authors should quantify the ratio of pAKT to total AKT and pAMPK to total as well. This would enrich the dataset examining these parameters and would allow for a more nuanced interpretation of the signaling data of these proteins.

The first sentence in the Discussion is a bit abrupt. Rephrasing in a way that transitions a bit smoother into the Discussion from the Results would enhance readability.

It is unfortunate that body composition data from NMR was not reported. Bodyweight changes were especially robust, but it would be of interest to determine if the resistance trained animals gained muscle mass and therefore were significantly leaner than the diabetic sedentary animals. The authors should include text that addresses this point as a limitation in the current study.

Author Response

Thank you for your comprehensive review. We provided a point-by-point response to your comments in a file below.

Response to Reviewer 2 Comments

The authors would like to thank for your comments on our work and for providing valuable feedback.

Point 1: For the exercise protocol, is there a way to express the load as a percentage of the animals’ maximal capacity? In humans, exercise intensity is usually prescribed on the basis of  a percentage of 1 repetition maximum. It would be of interest to know the intensity at which the animals were training.

Response 1: Thank you for your helpful comment. Effects of resistance exercise training intervention have been well known in animal studies. Generally, in many studies, exercise intensity adapted load of fixed rate based on body weight. Therefore, this study adapted the resistance training protocol from a previous study (Karimian et al. 2015). 

Point 2: The authors should change “improved” in reference to food intake in Figure 1B to “decreased”. This would make for a more straightforward interpretation of the data.

Response 2: Thank you for your feedback. We have corrected it to ‘decreased’ in the revised manuscript (page 4, lines 154)

Point 3: The authors show in figure 2 that fasting glucose in the trained animals was partially restored to a level close to the controls. However, insulin also was altered by the treatment. Given that both fasting glucose and insulin are important to understanding the whole-body glucose homeostasis of the animals, the authors should include a panel that incorporates both measures in an index such as HOMA-IR or QUICKI, which has been shown to correlate well with insulin clamp studies.

Response 3: We appreciate your kind review and totally agree with the reviewer about that. In many studies, it has been demonstrated that exercise intervention improves insulin resistance in a diabetic condition. In contrast, Hays et al. (2006) and Johnson et al. (2009) reported that HOMA-IR assessment does not always track significant changes in insulin sensitivity and regular exercise had no influence on HOMA-IR index. Similarly, in our study, HOMA-IR levels had no significant differences between all groups although significant changes in insulin and glucose levels were shown. In our study, increased insulin secretion is considered effects through exercise-induced GLP-1R activation, regardless of insulin resistance. Therefore, we did not present HOMA-IR in manuscript. We ask for your understanding regarding this matter. Our study focused on mechanisms about insulin secretion and glycemic control of brain region induced by resistance exercise. We will conduct further study, considering your comments such as HOMA-IR or QUICKI.

Point 4: For the WB data on AKT and AMPK, the authors should quantify the ratio of pAKT to total AKT and pAMPK to total as well. This would enrich the dataset examining these parameters and would allow for a more nuanced interpretation of the signaling  data of these proteins.

Response 4: Thank you for your contractive and comprehensive review of paper. The following changes have been made in the Result section of the revised manuscript (Page 5 lines 160~165, 174; Page 6 lines181, 184).

Point 5: The first sentence in the Discussion is a bit abrupt. Rephrasing in a way that transitions a bit smoother into the Discussion from the Results would enhance readability.

Response 5: We accepted your suggestion and have added sentences as follows: “Incretin hormone-based therapies are commonly used to patients with T2DM to reduce blood glucose levels. However, it may require a more efficient and safer strategies for hyperglycemia without side effects.” (Page 6, lines 194-196).

Point 6: It is unfortunate that body composition data from NMR was not reported. Body weight changes were especially robust, but it would be of interest to determine if the resistance trained animals gained muscle mass and therefore were significantly leaner than the diabetic sedentary animals. The authors should include text that addresses this point as a limitation in the current study.

Response 6: Thank you for your helpful comment. We have added a limitation in the revised manuscript as follows: “However, this current study has a limitation as follows; we did not measure body composition except body weight, adipose tissue, lipid profile, and muscle mass.”. If possible, we will reflect these recommended factors in further study (Page 7, lines 256-257).

Reference

1. Karimian, J. Khazaei, M. Shekarchizadeh, P. Effect of Resistance Training on Capillary Density Around Slow and Fast Twitch Muscle Fibers in Diabetic and Normal Rats. Asian J Sports Med 2015, 6, (4), e24040.

2. Hays NP, Starling RD, Sullivan DH, Fluckey JD, Coker RH, Evans WJ. Comparison of insulin sensitivity assessment indices with euglycemic-hyperinsulinemic clamp data after a dietary and exercise intervention in older adults. Metabolism. 2006, 55(4):525-532.

3. Johnson NA, Sachinwalla T, Walton DW, Smith K, Armstrong A, Thompson MW, George J. Aerobic exercise training reduces hepatic and visceral lipids in obese individuals without weight loss. Hepatology. 2009 50(4):1105-12.

We have shown our responses to your comments in red in the revised manuscript.